

# The application of the propensity score matching method in stock prediction among stocks within the same industry

Shengnan Li and Lei Xue

School of Communication and Information Engineering, Shanghai University, Shanghai, China

## ABSTRACT

Stock price prediction is crucial in stock market research, yet existing models often overlook interdependencies among stocks in the same industry, treating them as independent entities. Recognizing and accounting for these interdependencies is essential for precise predictions. Propensity score matching (PSM), a statistical method for balancing individuals between groups and improving causal inferences, has not been extensively applied in stock interdependence investigations. Our study addresses this gap by introducing PSM to examine interdependence among pharmaceutical industry stocks for stock price prediction. Additionally, our research integrates Improved particle swarm optimization (IPSO) with long short-term memory (LSTM) networks to enhance parameter selection, improving overall predictive accuracy. The dataset includes price data for all pharmaceutical industry stocks in 2022, categorized into chemical pharmaceuticals, biopharmaceuticals, and traditional Chinese medicine. Using Stata, we identify significantly correlated stocks within each sub-industry through average treatment effect on the treated (ATT) values. Incorporating PSM, we match five target stocks per sub-industry with all stocks in their respective categories, merging target stock data with weighted data from non-target stocks for validation in the IPSO-LSTM model. Our findings demonstrate that including non-target stock data from the same sub-industry through PSM significantly improves predictive accuracy, highlighting its positive impact on stock price prediction. This study pioneers PSM's use in studying stock interdependence, conducts an in-depth exploration of effects within the pharmaceutical industry, and applies the IPSO optimization algorithm to enhance LSTM network performance, providing a fresh perspective on stock price prediction research.

# INTRODUCTION

The stock market is an indispensable component of the modern socio-economic landscape and serves as a crucial avenue for contemporary businesses to raise capital. The price of stocks, formed during public market transactions, is subject to fluctuations influenced by market supply and demand dynamics. Simultaneously, certain real-world factors in the financial markets, such as market sentiment, macroeconomic elements, and geopolitical

Corresponding author
Lei Xue, 16301098@163.com

events, also exert influence on the stock market (*Babaei, Hübner & Muller, 2023*; *Yang & Yang, 2021*). Stock price prediction stands as a pivotal issue in stock market research, with various forecasting models applied to anticipate stock price movements (*Nikou et al., 2019*; *Obthong et al., 2020*). However, the majority of current stock price prediction models treat each stock as an independent entity, utilizing its individual historical information to forecast price trends, thereby overlooking potential interdependencies among different stocks. The phenomenon of stock interdependence refers to the manifestation of high correlation and similar stock price fluctuation patterns in the stock market over a specific period. In reality, the prices of related stocks can significantly impact each other, especially among companies in the same industry (*King, 1966*). These companies often exhibit strong correlations in both daily and long-term price movements. Effectively identifying interdependencies among stocks can aid researchers in enhancing the accuracy of stock price predictions.

Previous research extensively used various quantitative methods, including Granger causality models and vector autoregressive frameworks, to study stock interdependence. However, these investigations often faced observational data challenges, such as selective biases in stock choices and confounding factors. Selective biases could arise from preferences based on market value, industry, and corporate governance. Simultaneously, confounding factors encompass market sentiment and macroeconomic indicators, influencing both stock selection and observed interdependence effects. Propensity Score Matching (PSM), introduced by Paul Rosenbaum and Donald Rubin in 1983, addresses these challenges (*Li et al., 2022*). It is a statistical method designed to balance individuals between groups by matching them on key characteristics, reducing biases in observational data. PSM strengthens comparisons and enhances causal inferences by mitigating the impact of confounding factors and selective biases. Despite this, there's a notable absence of studies applying PSM to stock interdependence investigations.

Recently, long short-term memory (LSTM) networks have become a staple in stock price prediction, frequently used as components in encoder–decoder networks for time series forecasting (*Van Houdt, Mosquera & Nápoles, 2020*). Recognized for their proficiency in capturing sequential dependencies, LSTM networks also combat issues like gradient vanishing and exploding, presenting a robust option for time series prediction models (*Alom et al., 2019*). However, the effectiveness of LSTM networks relies on parameter selection, often guided by user experience, impacting predictive performance. To tackle this, researchers propose metaheuristic optimization algorithms like simulated annealing, genetic algorithms, and particle swarm optimization (PSO) for parameter optimization. *Ceylan (2021)* utilized PSO to enhance prediction accuracy in the grey model, while *Huang et al. (2020)* employed improved particle swarm optimization (IPSO) to boost overall model performance by adjusting backpropagation parameters. PSO stands out for its computational efficiency, swiftly converging to satisfactory results in complex optimization problems. The algorithm leverages individual and collective information for collaborative searching, enhancing effectiveness by adjusting parameters like inertia weight, learning factor, population, velocity, and displacement.

This study is dedicated to addressing prevalent issues in current stock price prediction methods, namely incomplete information and insufficient stock relationships. Through the application of PSM, our research delves into the interdependence among stocks, aiming to mitigate data selection biases and reveal meaningful relationships within the same industry. Simultaneously, we employ IPSO to optimize the LSTM model by adjusting the number of neurons, learning rate, and iteration count, resulting in the creation of the IPSO-LSTM model. Additionally, we integrate the interdependence effects among stocks into stock price prediction to enhance overall predictive accuracy. Our research findings demonstrate that incorporating non-target stock data from the same sub-industry through PSM significantly improves the predictive accuracy of target stock prices. This underscores the positive impact of PSM in stock price prediction and its potential to enhance overall predictive accuracy.

The primary contributions of this article are threefold: (1) pioneering the use of PSM to study the interdependence among stocks and incorporating this relationship into stock price prediction; (2) conducting an in-depth exploration of interdependence effects among stocks within the pharmaceutical industry, offering insights for subsequent studies in the same industry; (3) in the predictive research section, applying the IPSO optimization algorithm to address hyperparameter issues in the LSTM network, thereby improving stock price prediction accuracy.

The remaining sections of the article are structured as follows: The "Literature Review" section summarizes recent studies on interdependencies, PSM application, and stock price forecasting. In the "Materials and Methods" section, we introduce the PSM methodology, IPSO algorithm and our study design. The "Results" section presents matched experimental data, model results, and predictions, analyzed with evaluation indicators. The "Discussion and Conclusion" section summarizes the effectiveness of the PSM method in stock price prediction.

# LITERATURE REVIEW

In recent years, heightened attention to stock market dynamics has been fueled by the intricate complexities and inherent interdependencies within the market. Recognizing the interconnected nature of stocks within the same industry is now crucial for precision in stock price prediction and informed investment decisions.

## The impact of real-world factors in the financial markets on the stock market

As we delve deeper into the study of stock markets, it becomes crucial to focus on a number of real-world factors. *Babaei, Hübner & Muller (2023)* find that changes in certain financial risk factors, economic policy uncertainty (EPU), and global geopolitical risk (GPR) significantly affect the interdependence of stock markets in the G7 member countries. *Aladesanmi, Casalin & Metcalf (2019)* point out that conditional correlations between the EPU of the US and the UK have an impact on their respective stock markets. *Belhoula, Mensi & Naoui (2023)* highlight the important relationship between stock market efficiency, crude oil prices, and the COVID-19 outbreak. Additionally, *Khan et al. (2020)*

found that public sentiment and political situations have a significant impact on the accuracy of machine learning algorithms in predicting stock market trends. *Yang & Yang (2021)* confirmed that stock market predictions are enhanced in the presence of escalating geopolitical risks.

## Interdependence among stocks

Several studies underscore the existence of interdependence effects among stocks, emphasizing their significance in financial modeling (*Atahau, Robiyanto & Huruta, 2022*; *Bai, Cui & Zhang, 2019*; *Fang et al., 2018*; *Huang & Wang, 2018*; *Kumar, Singh & Jain, 2023*; *Lee, 2021*; *Li & Wei, 2018*; *Ma et al., 2022*; *Ma et al., 2019*; *Nasreen et al., 2020*; *Niu, 2021*; *Wang et al., 2018*; *Wang & Chong, 2018*; *Wang et al., 2019*; *Wu et al., 2021*; *Xu et al., 2017*; *Zhao et al., 2022*; *Zhou et al., 2023*). For instance, *Ma et al. (2019)* investigated market linkage between Shanghai and Hong Kong. *Bai, Cui & Zhang (2019)* applied K-means clustering to Shanghai A-shares, identifying stocks with similar patterns. *Zhou et al. (2023)* explored mutual dependence of tail risk among Chinese stock market sectors, highlighting crucial channels for risk contagion in closely tied industrial chains. *Wang & Chong (2018)* conducted integrated tests on stock interconnectivity between mainland China and Hong Kong, revealing cointegration of A-shares and H-shares after the Shanghai-Hong Kong Stock Connect program introduction. Stocks within the same industry often exhibit co-movements influenced by macroeconomic factors, market trends, and industry-specific events. Recognizing and quantifying these interdependencies are essential for robust predictive models.

## Propensity score matching (PSM) applications

PSM has proven invaluable in diverse fields, addressing biases and enhancing research reliability (*Caliendo & Kopeinig, 2008*; *Kane et al., 2020*). *Su (2021)* investigated the return performance of green investments using a multifactor model and PSM, revealing underperformance and lower resilience against extreme downside risks compared to conventional stocks. *Guo, Yang & Fan (2022)* explored corporate social responsibility disclosure impact on stock price informativeness in China, employing PSM and a difference-in-difference approach. *Garg et al. (2022)* used PSM with two-stage least squares to examine whether management quality mitigates the positive correlation between corporate tax avoidance and firm-specific stock price plunge risk. Despite success in diverse applications, the potential of PSM in uncovering intricate interdependence among stocks within the same industry remains largely untapped.

## Current state of stock prediction

Despite the sophistication of current stock prediction methods, they often fall short in capturing the complete spectrum of interdependence among stocks. Extensive research indicates that LSTM models excel in effectively extracting time series information, demonstrating remarkable performance in stock price prediction (*Babu et al., 2023*). *Fischer & Krauss (2018)* conducted a comprehensive analysis, comparing LSTM against various classifiers, highlighting its pronounced superiority. *Kim & Won (2018)* innovatively

integrated LSTM with GARCH models, revealing consistent outperformance across various metrics in a hybrid model.

In the pursuit of optimization, IPSO plays a crucial role in fine-tuning LSTM parameters, such as the number of neurons, learning rate, and iteration count (*Jovanovic et al., 2022*; *Stankovic et al., 2022*; *Suddle & Bashir, 2022*). *Lai & Wang (2023)* successfully applied IPSO to enhance the accuracy of short-term passenger flow prediction in rail transit using LSTM neural networks, showcasing feasibility and efficacy. Similarly the IPSO-LSTM public opinion prediction model of *Mu et al. (2023)* significantly enhanced the accuracy of public opinion trend prediction. *Ji, Liew & Yang (2021)* introduced an IPSO-LSTM model for stock price prediction, demonstrating its superiority over relevant baseline models on the Australian stock market index, including support vector regression, pure LSTM, and PSO-LSTM.

Based on the existing literature summary, we have identified research gaps. Firstly, there is a gap in recognizing and incorporating interdependencies among stocks within the same industry in stock price prediction models. Secondly, we found that PSM is underutilized in stock market research, particularly for studying interdependence among stocks. Our study pioneers the use of PSM in this context, providing a new perspective on stock market dynamics. Thirdly, while literature acknowledges the importance of parameter optimization in machine learning models, there is a potential gap in exploring and applying IPSO in conjunction with LSTM networks for stock price prediction within specific industries. Our study contributes by integrating these methods to enhance predictive accuracy.

In summary, our research addresses identified gaps by introducing PSM to study stock interdependence, emphasizing its underutilization, and applying IPSO to optimize parameters in LSTM networks for improved stock price predictions within the pharmaceutical industry.

## MATERIALS & METHODS

### Study of interconnectivity—PSM method

We delve into the interdependence between stocks by applying PSM, aiming to mitigate data selection bias and reveal meaningful relationships within the same industry.

### The principle of matching

The matching method offers an intuitive approach to estimate the treatment effect by identifying untreated individuals with similar observable characteristics to those who underwent treatment. Direct matching, however, faces challenges when dealing with multiple features. In empirical research, direct matching is not commonly employed due to its complexity.

### The principle of PSM

To address the complexity of direct matching, especially with continuous variables, Rosenbaum and Rubin introduced the PSM method. The core concept of PSM involves converting the multi-dimensional variable $X$ into a one-dimensional propensity score $ps(X_i)$ using functional relationships. Subsequent matching is then carried out based on this propensity score.

The propensity score (*Rickles, 2016*) represents the probability of individuals with observable features $X_i = x$ accepting treatment, denoted as:

$$ps(X_i = x) = P(D_i = 1|X_i = x) \tag{1}$$

In this Eq. (1), $P$ denotes the conditional probability. $X_i$ represents the multidimensional matching variables for the control group, specifically referring in this context to the stock data's opening price, highest price, and lowest price.

## Latent outcomes

If individual $i$ undergoes a certain treatment action denoted as $D_i$, its corresponding outcome is denoted as $Y_i(1)$; if it does not undergo the treatment, the outcome is denoted as $Y_i(0)$. These two outcomes are referred to as latent outcomes, as shown in Eq. (2):

$$latent\ outcomes = \begin{cases} Y_i(0), & \text{if } D_i = 0 \\ Y_i(1), & \text{if } D_i = 1 \end{cases} \tag{2}$$

where $D_i = 0$ indicates that individual $i$ did not undergo the treatment, while $D_i = 1$ indicates that individual $i$ received the treatment. It is referred to as a latent outcome because these two outcomes inherently exist within individual $i$, but may not necessarily manifest; if they don't manifest, we cannot observe them.

In this experiment, $i$ refers to the stock number. The treatment action $D_i$ represents whether other stocks in the same industry serve as the treatment. When the stock is a non-target stock in the same industry, $D_i = 1$; when the stock is the target stock, $D_i = 0$. The latent outcome $Y_i$ represents the closing price of the target stock.

## Treatment effect

The treatment effect of the treatment action $D_i$ on individual $i$ is the difference between the potential outcomes $Y_i(1)$ and $Y_i(0)$ for individual $i$ who received the treatment ($D_i = 1$) and did not receive the treatment ($D_i = 0$), respectively. As shown in Eq. (3):

$$\gamma_i = Y_i(1) - Y_i(0). \tag{3}$$

As Eq. (3) indicates, the treatment effect represents the causal utility of the treatment action $D_i$ on outcome $Y_i$.

The treatment effect of a treatment action may vary among different individuals, indicating heterogeneity in individual treatment effects. Therefore, when measuring treatment effects, we typically use a simple statistic to describe the average effect, known as the average treatment effect. For different groups, we can define different average treatment effects. The average treatment effect on the treated (ATT) refers to the average treatment effect on individuals who receive the treatment. Typically, ATT is the outcome of utmost concern, as it represents the direct consequence of the treatment intervention. The formal definition of the ATT is the expected value of the treatment effect for all individuals who receive the treatment. As shown in Eq. (4):

$$ATT = E(Y_i(1)|D_i = 1, X_i = x) - E(Y_i(0)|D_i = 1, X_i = x). \tag{4}$$

The PSM method satisfies the assumption of conditional independence, wherein, given the observable features, the latent outcomes are independent of the treatment status. Its mathematical expression is shown in Eq. (5):

$$[Y_i(1), Y_i(0)] \perp D_i | X_i. \tag{5}$$

In this research experiment, ATT represents the impact of non-target stocks in the same industry on the closing price of the target stock. As it satisfies Eq. (5), ATT can also be expressed as:

$$ATT = E(Y_i(1)|D_i = 1, X_i = x) - E(Y_i(0)|D_i = 0, X_i = x). \tag{6}$$

In other words, after PSM controls for observable characteristics, the ATT represents the expected value of non-target stock B over the closing price of target stock A in the same industry, minus the expected value of target stock A over its own closing price. If the ATT surpasses 1.96 (for propensity score significance, 1.96 is commonly used as the significance level), it indicates that non-target stock B within the same industry does indeed have a significant effect on the closing price of target stock A.

## Prediction and validation—IPSO-LSTM model

This study aims to validate the practical significance of the PSM method in exploring interdependencies among stocks and enhancing the accuracy of stock price prediction. The IPSO-LSTM model is employed to investigate whether the PSM method effectively uncovers relationships between stocks within the same industry.

## The principle of PSO

In the context of PSO, the optimization problem's latent solution is conceptualized as a singular particle that continually explores the search space. This particle adjusts its position based on its own experience and the experiences of the best individuals during the quest for the optimal location. The PSO process initiates by randomly generating a set of solutions and iterates to seek the optimal solution by tracking the particle exhibiting the best performance in the current space. Within the multi-dimensional search space, a group consists of "m" particles. In the "t"-th iteration, the position and velocity of the "i"-th particle are represented as $X_{i,t}$ and $V_{i,t}$, respectively. Each particle updates its own position and velocity by monitoring two optimal solutions. The first is the best solution the particle has found so far, known as the individual extremum ($pbest_i$). The second is the current best solution discovered by the entire population, known as the global best solution ($gbest_t$). While searching for these two optimal solutions, particles adjust their velocity and new position using the following formula:

$$V_{i,t+1} = w * V_{i,t} + c_1 * rand * (pbest_i - X_{i,t}) + c_2 * rand * (g\,best_t - X_{i,t}) \tag{7}$$

$$X_{i,t+1} = X_{i,t} + \lambda * V_{i,t+1} \tag{8}$$

In the formula, $w$ stands for the inertia weight, $c_1$ and $c_2$ represent the learning factors, *rand* denotes a random number between 0 and 1, $\lambda$ serves as the velocity coefficient, where $\lambda$ equals 1.

## The principle of IPSO

To enhance the global optimization ability and convergence speed of the basic PSO, the IPSO method suggests the use of a nonlinearly changing inertia weight. In the basic PSO, a fixed inertia $w$ can weaken the algorithm's global optimization capability and impede its convergence speed. In this study, we begin by modifying the inertia $w$ according to the formulation Eq. (9):

$$w = w_{max} - (w_{max} - w_{min}) * \arcsin \frac{t}{t_{max}} * \frac{2}{\pi} \tag{9}$$

In the formula, $w_{max}$ and $w_{min}$ are the maximum and minimum values of $w$, respectively. $t$ is the current iteration number. $t_{max}$ is the maximum number of iterations. When $t$ is small, $w$ approaches $w_{max}$, and the decrease rate of $w$ is slow, ensuring the algorithm's global optimization ability. As $t$ increases, $w$ decreases nonlinearly, and the decrease rate of $w$ increases rapidly, ensuring the algorithm's local optimization ability. This allows the algorithm to flexibly adjust its global and local optimization abilities.

The second enhancement encompasses the integration of the mutation operation from genetic algorithms into the fundamental PSO framework, leading to the introduction of adaptive mutation. As evolutionary generations progress, the probability of mutation diminishes, aiding particles in mitigating the likelihood of becoming ensnared in local optima. Equation (10) for calculating adaptive mutation probability is as follows:

$$rand > \frac{1}{2}(1 + \frac{t}{t_{max}}) \tag{10}$$

As the value of $t$ increases, the right side of the inequality gradually expands within the range of (0.5, 1). In this context, $rand$ generates a random number within the interval of (0, 1). Owing to the incremental nature of the right side of the inequality, the probability of $rand$ surpassing this threshold dwindles. As a result, the probability of particle mutation diminishes.

## Optimizing LSTM using IPSO

**Step 1**: Parameter initialization. Begin by setting the population size, number of iterations, learning factors, and the range of values for positions and velocities.

**Step 2**: Initialization of particle positions and velocities. Generate a population particle $X_{i,0}(h_1, h_2, \varepsilon, n)$ through random selection. Here, $h_1$ denotes the number of neurons in the first hidden layer, $h_2$ represents the number of neurons in the second hidden layer, $\varepsilon$ signifies the learning rate of LSTM, and $n$ indicates the number of iterations for LSTM.

**Step 3**: Definition of particle evaluation function. Assign the particle $X_{i,0}$ from Step 2 to the parameters of the LSTM. Divide the data into training samples and test samples. Input the training samples for neural network training. Upon reaching the iteration limit, derive the training sample output value $\widehat{y_{train}}$ and the test sample output value $\widehat{y_{test}}$ of the neural network. Therefore, the fitness value $fit_i$ of individual $X_i$ is defined as follows:

$$fit_i = \frac{1}{n} \sum_{i=1}^{n} (y_{test} - \widehat{y_{test}})^2 \tag{11}$$

In the Eq. (11), $y_{test}$ represents the true value of the test sample, and $\widehat{y_{test}}$ represents the predicted value of the test sample.

**Step 4**: Calculate fitness and optimal positions. Calculate the fitness value for each particle's position. Determine both individual and global optima using the initial particle fitness values. Assign each particle's best position as its historical best position.

**Step 5**: Iteration update. In each iteration, adjust the particle's velocity and position using Eqs. (7) and (8), employing the individual and global optima. Compute the fitness value for the new particle's position and update the individual and global optima with the fitness values from the new particle population.

**Step 6**: PSO algorithm termination. Upon reaching the PSO algorithm's maximum iteration limit, input the prediction data into the LSTM model trained using the optimal particle. This will yield the projected closing price for the target stock.

## Experimental design

This study employs the PSM method and utilizes a 1:1 matching approach to pair target stocks (treatment group) with control group samples from the same industry (*i.e.*, non-target stocks in the same industry). The objective is to analyze the impact of non-target stock data from the same industry on the closing price of the target stocks, as represented by the ATT value. A statistically significant ATT value will prompt further validation using the IPSO-LSTM model.

During the matching process, each target stock undergoes individual pairing with non-target stocks from the same industry. Stata is employed to calculate ATT values, identifying non-target stocks displaying significant interdependence with the target stock. The average propensity scores of these matched non-target stocks are computed and serve as weighting values in the IPSO-LSTM model.

For the prediction and validation phase, the data of the target stock and the non-target stocks demonstrating significant interdependence are merged into a new dataset. The IPSO-LSTM model is then utilized to predict closing prices for the target stock. This step aims to assess whether incorporating data from non-target stocks with notable interdependence enhances the accuracy of closing price predictions for the target stock.

The analysis employs Stata 17 and Python 3.9 as the primary tools for research and experimentation.

## Experimental data

In this study, daily data for all stocks in the pharmaceutical industry from the year 2022 is selected from the TongDaXin database, totaling 296 stocks. These 296 stocks are further categorized into three sub-industries: 146 stocks in chemical pharmaceuticals, 75 stocks in biopharmaceuticals, and 75 stocks in traditional Chinese medicine. The pharmaceutical industry is a crucial sector in the global economy and plays a vital role in healthcare. This research will conduct analyses on these three sub-industries and select the top five companies based on market capitalization from each sub-industry as the target stocks for the study.

**Table 1 Definition of variables in the propensity score matching method.**

| Variable type | Variable name | Definition |
|---|---|---|
| Dependent variables | The closing price of the target stock. | The closing price of the top five ranked stocks by market capitalization in each sub-industry serves as the dependent variable. |
| Core explanatory variables | Whether it pertains to the target stock. | Once the target stock is selected, it is assigned a value of 0, while other stocks in the same sub-industry are assigned a value of 1, studying the impact of 1 on 0. |
| Controlled variables | The opening price. | To control for the opening price variable, the opening prices of all non-target stock companies within the same sub-industry are matched one-to-one with the opening price of the target stock company over time. |
| | The highest price. | To control for the highest price variable, the highest prices of all non-target stock companies within the same sub-industry are matched one-to-one with the highest price of the target stock company over time. |
| | The lowest price. | To control for the lowest price variable, the lowest prices of all non-target stock companies within the same sub-industry are matched one-to-one with the lowest price of the target stock company over time. |

## PSM data

In this study, the closing price of the top five companies by market capitalization in each sub-industry is used as the dependent variable. The core explanatory variable is whether a stock is a target stock. Once the target stocks are identified, they are assigned a value of 0, while other stocks within the same sub-industry are assigned a value of 1, thereby investigating the impact of 1 on 0. To enhance the precision and validity of PSM, control variables should encompass factors that simultaneously influence both the dependent variable and the core explanatory variable. This study selects the opening price, highest price, and lowest price as control variables. Specific variables are described in Table 1.

## IPSO-LSTM model data

In the process of IPSO-LSTM prediction and validation using Python, the stock data utilized comprises the opening price, closing price, highest price, and lowest price. These data are sourced from the TongDaXin database and are partitioned into 80% training data and 20% testing data. This division ensures a robust training of the model on a substantial portion of the data, followed by rigorous testing on a separate set to evaluate the model's predictive performance and generalization capabilities. This approach aims to enhance the reliability and accuracy of the IPSO-LSTM model in forecasting stock prices based on the specified features.

## Evaluation indicators of the experiment

In the PSM evaluation process, two critical assumptions are considered: the balance assumption and the common support assumption. The balance assumption, which requires no significant differences between the treatment and control groups after matching, is assessed using standardized bias (%bias) and the $T$-test of mean differences. The common

support assumption, ensuring that score values for both groups fall within a shared range, is verified through visual examination using the "psgraph" command in Stata.

For regression models, evaluation criteria following *Gülmez (2022)* guidelines are employed. These include root mean square error (*RMSE*), mean absolute error (*MAE*), mean absolute percentage error (*MAPE*), and the coefficient of determination ($R^2$). Lower values for *RMSE*, *MAE*, and *MAPE* indicate better model performance, while an $R^2$ value closer to 1 suggests a superior model fit. These metrics collectively gauge the accuracy and effectiveness of the model's predictions based on input data. The computation formulas for these criteria are as follows:

$$MAPE = \frac{100\%}{n} \sum_{i=1}^{n} |\frac{y_i - \hat{y}_i}{y_i}| \tag{12}$$

$$RMSE = \sqrt{\frac{1}{n} \sum_{i=1}^{n} (y_i - \hat{y}_i)^2} \tag{13}$$

$$MAE = \frac{1}{n} \sum_{i=1}^{n} |y_i - \hat{y}_i| \tag{14}$$

$$R^2 = 1 - \frac{\sum (y_i - \hat{y}_i)^2}{\sum (y_i - \overline{y}_i)^2} \tag{15}$$

**Model comparison experiment**

To ensure research robustness and comprehensive result validation, this article introduces another analytical approach—ridge regression analysis—for comparison. To comprehensively evaluate the comparison between propensity score matching and ridge regression analysis, this study introduces the baseline LSTM model, with parameters unoptimized through IPSO. The baseline LSTM model allows for a direct comparison of the effectiveness of propensity score matching and ridge regression analysis in studying interdependence without additional optimization. This ensures a more comprehensive and objective assessment of the two methods. Specific analytical methods can be found in Appendix 2.

## RESULTS

This study focuses on forecasting closing prices in the chemical pharmaceutical industry. Results for other sub-industries are available in Appendix 1 (Figs. S1–S7, Tables S1–S8). Focusing on the chemical pharmaceutical sector, our analysis zeroes in on the top five companies, ranked by market capitalization. These companies include Wuxi Apptec Co.,Ltd (stock code: 603259.sh), Jiangsu Hengrui Pharmaceuticals Co., Ltd (stock code:

600276.sh, referred to as "Hengrui" hereafter), Shanghai Fosun Pharmaceutical Co., Ltd (stock code: 600196.sh, referred to as "Fuxing" hereafter), Humanwell Healthcare Co., Ltd (stock code: 600079.sh, referred to as "Renfu" hereafter), and Zhejiang Huahai Pharmaceutical Co., Ltd (stock code: 600521.sh, referred to as "Huahai" hereafter).

Utilizing the 1:1 nearest neighbor matching technique and controlling for opening, highest, and lowest prices of target stocks in the chemical pharmaceutical industry against non-target stocks, and employing the Stata tool, we have successfully identified non-target stock companies within the pharmaceutical sector that demonstrate a significant ATT and have successfully passed the PSM test. The identified companies are Hunan Warrant Pharmaceutical Co., Ltd (stock code: 688799.sh, referred to as "Huana" hereafter) and Hitgen Inc (stock code: 688222.sh, referred to as "Chengdu" hereafter). The resulting stock portfolios are as follows: Hengrui-Huana, Fuxing-Hengrui, Fuxing-Huana, Renfu-Chengdu, and Huahai-Renfu.

Table 2 presents the results of the balance hypothesis tests for the five sets of matched stocks. The standardized bias (%bias) across these sets is consistently below 10%, indicating a notable balance effect and suggesting no significant differences among the matched variables post-matching. Additionally, the inter-group mean $T$-test results reveal no significant differences between the stocks after matching, further confirming the fulfillment of the balance hypothesis (Table 2).

The outcomes of the common support tests for the five sets of stock data are visually presented in Fig. 1 and quantitatively detailed in Table 3. Figure 1 distinctly illustrates that a substantial majority of data points from both the control and treatment groups are situated within the common value range. Table 3 further provides specific data from the common support domain test. For instance, considering the daily stock data samples from 2022, the Huahai-Renfu stock had a total of 480 data points, with 470 falling within the common range, and only 10 data points lying outside this interval. The combined evidence affirms that all five sets of stocks successfully adhere to the common support assumption, underscoring favorable matching outcomes.

Table 4 displays the weight values derived by averaging the propensity score values of each successfully matched non-target stock within the common support domain. Across all five sets of stock matching results in Table 4, the ATT values consistently surpass 1.96. This suggests that the disparities observed between the treatment and control groups are unlikely to be attributed to random chance. These results indicate a significant and non-random impact of the matched stocks on the closing prices of the target stocks. Notably, the highest ATT value is observed for Fuxing-Huana at 9.51, underscoring a particularly pronounced influence in this specific pairing.

Figure 2 illustrates the comparative analysis of the IPSO-LSTM model for predicting Hengrui's stock price. The orange line represents the true values of Hengrui's stock in the test set. Predicted results without considering stock interdependence are depicted by the blue line, while the predicted results with stock interdependence taken into account are shown by the black line. Clearly, a notable enhancement in the forecasted results for Hengrui's stock price when Huana's stock data is taken into account. Further insights from Table 5, analyzing the evaluation metrics, reveal that accounting for interdependence

**Table 2  Results of balanced hypothesis testing for five stock data pairs in the chemical-pharmaceutical subsector.**

| Stocks | Variable | Unmatched Matched | Mean | | %Bias | %reduct Bias | T-test | | V(T)/ V(C) |
|---|---|---|---|---|---|---|---|---|---|
| | | | Treated | Control | | | T | P >\|T\| | |
| Hengrui-Huana | | U | 34.208 | 36.971 | −67.2 | | −7.37 | 0.000 | 0.53* |
| | open | M | 34.234 | 34.094 | 3.4 | 94.9 | 0.45 | 0.652 | 1.06 |
| | | U | 34.859 | 37.616 | −65.9 | | −7.22 | 0.000 | 0.60* |
| | high | M | 34.888 | 34.804 | 2.0 | 97.0 | 0.26 | 0.796 | 1.10 |
| | | U | 33.593 | 36.419 | −71.2 | | −7.80 | 0.000 | 0.49* |
| | low | M | 33.63 | 33.549 | 2.0 | 97.1 | 0.28 | 0.781 | 1.04 |
| Fuxing-Hengrui | | U | 36.971 | 41.619 | −85.2 | | −9.33 | 0.000 | 0.59* |
| | open | M | 37.282 | 37.093 | 3.5 | 95.9 | 0.43 | 0.666 | 0.86 |
| | | U | 37.616 | 42.51 | −87.7 | | −9.60 | 0.000 | 0.54* |
| | high | M | 37.936 | 37.787 | 2.7 | 96.9 | 0.35 | 0.729 | 0.86 |
| | | U | 36.419 | 40.781 | −82.4 | | −9.03 | 0.000 | 0.61* |
| | low | M | 36.717 | 36.528 | 3.6 | 95.6 | 0.44 | 0.661 | 0.82 |
| Fuxing-Huana | | U | 34.208 | 41.619 | −149.4 | | −1636 | 0.000 | 0.31* |
| | open | M | 34.208 | 34.269 | −1.2 | 99.2 | −0.19 | 0.846 | 0.99 |
| | | U | 34.859 | 42.51 | −147.8 | | −1619 | 0.000 | 0.32* |
| | high | M | 34.859 | 35.043 | −3.6 | 97.6 | −0.56 | 0.576 | 1.03 |
| | | U | 33.593 | 40.781 | −151.1 | | −1655 | 0.000 | 0.30* |
| | low | M | 33.593 | 33.584 | 0.2 | 99.9 | 0.03 | 0.974 | 0.99 |
| Renfu-Chengdu | | U | 16.741 | 18.688 | −78.2 | | −8.56 | 0.000 | 0.88 |
| | open | M | 17.164 | 17.283 | −4.8 | 93.9 | −0.58 | 0.561 | 1.25 |
| | | U | 17.049 | 19.169 | −83.5 | | −9.14 | 0.000 | 0.83 |
| | high | M | 17.486 | 17.578 | −3.6 | 95.7 | −0.45 | 0.656 | 1.22 |
| | | U | 16.401 | 18.303 | −77.9 | | −8.53 | 0.000 | 0.90 |
| | low | M | 16.812 | 16.917 | −4.3 | 94.5 | −0.52 | 0.602 | 1.31 |
| Huahai-Renfu | | U | 18.688 | 20.563 | −81.0 | | −8.88 | 0.000 | 1.59* |
| | open | M | 18.789 | 18.799 | −0.4 | 99.5 | −0.04 | 0.965 | 0.98 |
| | | U | 19.169 | 21.103 | −80.0 | | −8.76 | 0.000 | 1.51* |
| | high | M | 19.265 | 19.439 | −7.2 | 91.0 | −0.73 | 0.464 | 1.04 |
| | | U | 18.303 | 20.109 | −80.7 | | −8.44 | 0.000 | 1.68* |
| | low | M | 18.402 | 18.385 | 0.7 | 99.1 | 0.07 | 0.942 | 1.04 |

Note.
An asterisk (*) signifies a remaining variance difference between the groups.

yields significant improvements. Specifically, after considering the interdependence effect, Hengrui's stock price forecasting experiences a substantial reduction of 42.37% in *MAPE*, 30.38% in *RMSE*, and 44.22% in *MAE*. Concurrently, the coefficient of $R^2$ increases by 1.57%, indicating an overall enhancement in the accuracy and reliability of the predictive model.

Figure 3 depicts stock price prediction outcomes for Fuxing in various scenarios. The orange line represents actual values in the Fuxing test set. The blue, red, and black lines show stock price predictions when Fuxing uses historical data alone, combines it with Hengrui stock data, and combines it with Huana stock data, respectively. The visual representation

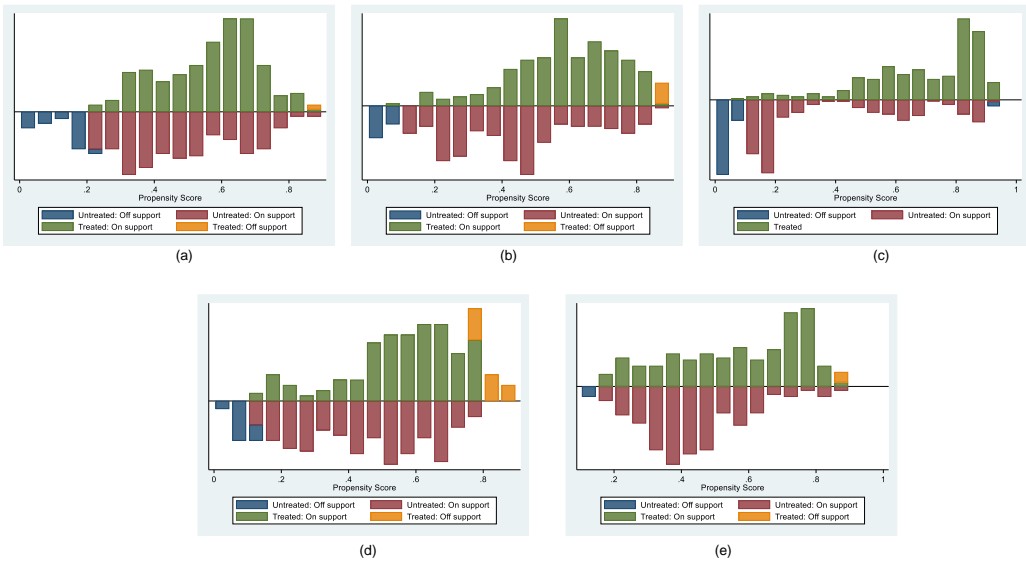

**Figure 1  Co-support Domain Visualization Results for Various Matchings in the chemical pharma-ceuticals subsector.** (A) Hengrui-Huana, (B) Fuxing-Hengrui, (C) Fuxing-Huana, (D) Renfu-Chengdu, and (E) Huahai-Renfu.

**Table 3  Results of the common support test for five pairs of stock data for the chemical-pharmaceutical subsector.**

| Stocks | psmatch2: Treatment assignment | psmatch2: Common support | | Total |
|---|---|---|---|---|
| | | Off support | On support | |
| Hengrui-Huana | Untreated | 33 | 207 | 240 |
| | Treated | 2 | 238 | 240 |
| | Total | 35 | 445 | 480 |
| Fuxing- Hengrui | Untreated | 22 | 218 | 240 |
| | Treated | 9 | 231 | 240 |
| | Total | 31 | 449 | 480 |
| Fuxing-Huana | Untreated | 63 | 177 | 240 |
| | Treated | 0 | 240 | 240 |
| | Total | 63 | 417 | 480 |
| Renfu-Chengdu | Untreated | 24 | 216 | 240 |
| | Treated | 28 | 212 | 240 |
| | Total | 52 | 428 | 480 |
| Huahai-Renfu | Untreated | 5 | 235 | 240 |
| | Treated | 5 | 235 | 240 |
| | Total | 10 | 470 | 480 |

**Table 4** Data portfolios of stocks in the chemical-pharmaceutical subsector with significant ATT and passing the PSM test.

| Stocks | ATT | Mean propensity score |
| --- | --- | --- |
| Hengrui-Huana | 5.55 | 0.53 |
| Fuxing-Hengrui | 5.44 | 0.51 |
| Fuxing-Huana | 9.95 | 0.56 |
| Renfu-Chengdu | 5.48 | 0.50 |
| Huahai-Renfu | 4.13 | 0.50 |

**Notes.**
ATT, Average Treatment Effect on the Treated; PSM, Propensity Score Matching.

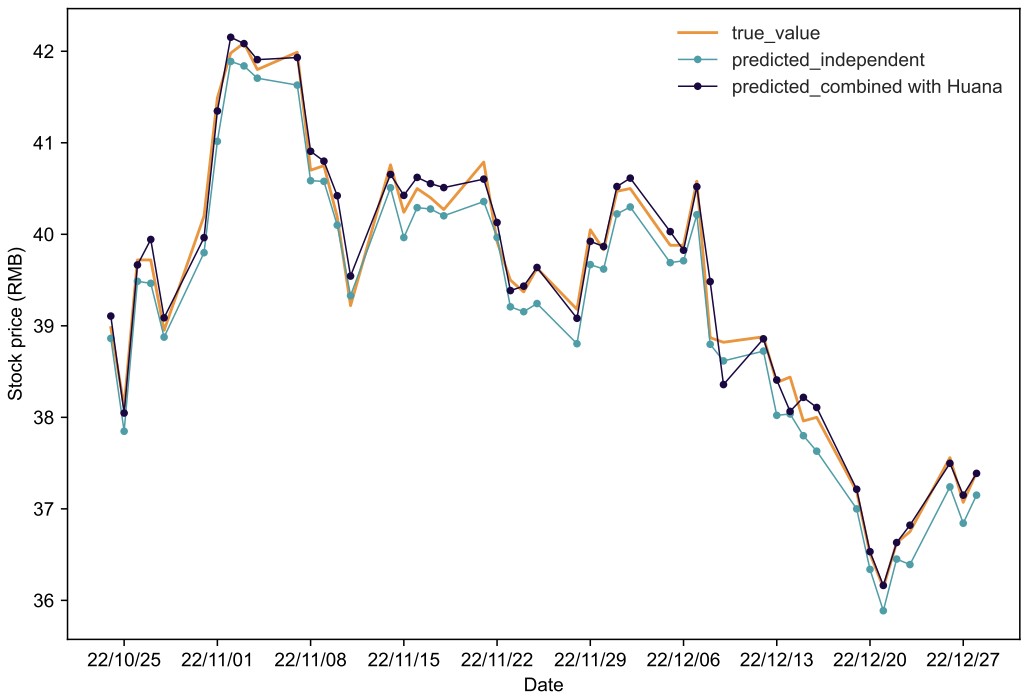

**Figure 2** **Comparative analysis of IPSO-LSTM model validation for Hengrui stock price forecasting with and without interdependence.** True value is depicted by the orange line, predicted results without stock interdependence are represented by the blue line, and predicted results with stock interdependence taken into account are illustrated by the black line.

in Fig. 3 clearly highlights the improvement in Fuxing's stock price predictions when incorporating data from both Hengrui and Huana.

Table 5 provides a detailed analysis of the evaluation metrics. Considering the interdependence effect, for Fuxing-Hengrui, there is a significant reduction of 41.86% in *MAPE*, 42.85% in *RMSE*, and 41.15% in *MAE*. Simultaneously, the coefficient of $R^2$ experiences a noteworthy increase of 2.59%. Similarly, for Fuxing-Huana, *MAPE*, *RMSE*, and *MAE* decrease by 53.49%, 45.4%, and 52.28%, respectively, while $R^2$ sees an increase of 2.7%. These results underscore the positive impact of considering interdependence on enhancing the accuracy and reliability of the stock price forecasting model for Fuxing.

**Table 5** Evaluation of IPSO-LSTM model prediction results, comparing target stock independent prediction with matching stock prediction.

| Stocks | Prediction type | MAPE | RMSE | MAE | $R^2$ |
|---|---|---|---|---|---|
| Hengrui-Huana | independent | 0.0059 | 0.2584 | 0.2388 | 0.9705 |
| | matching | 0.0034 | 0.1799 | 0.1332 | 0.9857 |
| Fuxing-Hengrui | independent | 0.0086 | 0.3729 | 0.3072 | 0.9630 |
| | matching | 0.0050 | 0.2131 | 0.1808 | 0.9879 |
| Fuxing-Huana | independent | 0.0086 | 0.3729 | 0.3072 | 0.9630 |
| | matching | 0.0040 | 0.2036 | 0.1466 | 0.9890 |
| Renfu-Chengdu | independent | 0.0054 | 0.1363 | 0.1182 | 0.9897 |
| | matching | 0.0036 | 0.1020 | 0.0757 | 0.9942 |
| Huahai-Renfu | independent | 0.0043 | 0.1292 | 0.0893 | 0.9782 |
| | matching | 0.0037 | 0.0955 | 0.0754 | 0.9881 |

**Notes.**
Mean absolute percentage error, MAPE; root mean square error, RMSE; mean absolute error, MAE; coefficient of determination, $R^2$.

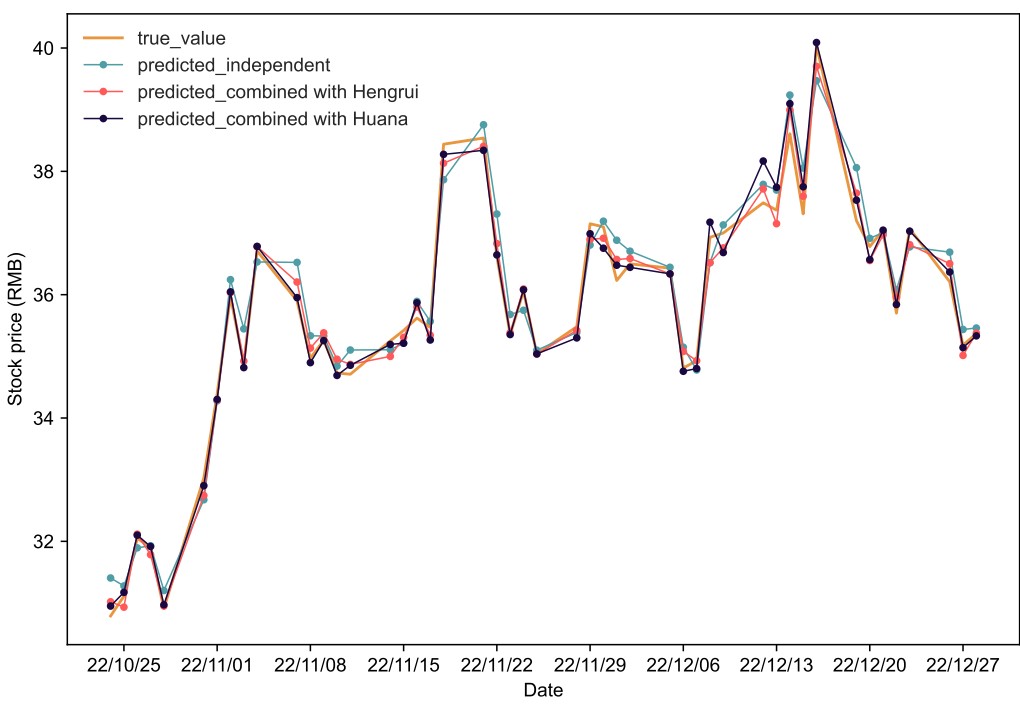

**Figure 3** **Comparative analysis of IPSO-LSTM model validation for Fuxing stock price forecasting with and without interdependence.** True value is depicted by the orange line, predicted results without stock interdependence are represented by the blue line, and predicted results with stock interdependence taken into account are illustrated by the red and black lines.

Figures 4 and 5 present the predictive outcomes for Renfu and Huahai, respectively. The corresponding evaluation metrics are detailed in Table 5. It is evident that incorporating matched stock data significantly enhances predictions compared to independent forecasts. The Renfu-Chengdu combination, when contrasted with independent predictions,

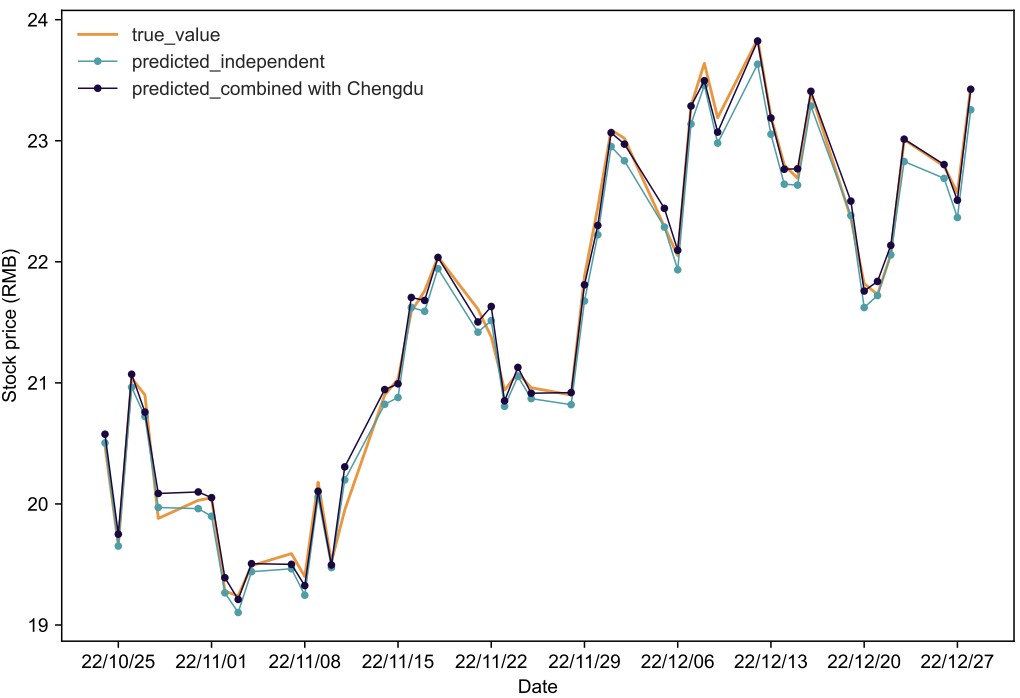

**Figure 4  Comparative analysis of IPSO-LSTM model validation for Renfu stock price forecasting with and without interdependence.** True value is depicted by the orange line, predicted results without stock interdependence are represented by the blue line, and predicted results with stock interdependence taken into account are illustrated by the black line.

showcases a notable decrease of 33.33% in _MAPE_, 25.17% in _RMSE_, and 35.96% in _MAE_, with an increase in $R^2$. Similarly, the Huahai-Renfu combination demonstrates a decrease of 13.95% in _MAPE_, 26.08% in _RMSE_, and 15.57% in _MAE_, coupled with an increase in $R^2$. These results underscore the positive impact of incorporating matched stock data on improving the precision and reliability of stock price predictions for both Renfu and Huahai.

The results of the comparative experiments for the chemical-pharmaceutical prediction model are in Appendix 2 (Figs. S8–S11 and Tables S9–S13). These findings underscore the advantages of propensity score matching in verifying stock interdependence. Simultaneously, they confirm that IPSO-LSTM excels in stock price prediction compared to the baseline LSTM.

## DISCUSSION AND CONCLUSION

This study employs causal theory, utilizing the PSM method, and validates its impact on stock price prediction through the IPSO-LSTM model. The dataset comprises 2022 price data for all pharmaceutical industry stocks, categorized into chemical pharmaceuticals, biopharmaceuticals, and traditional Chinese medicine sub-industries. Five target stocks were chosen for each sub-industry based on market capitalization and individually matched with all stocks in their respective sub-industries. Using Stata, ATT values identified

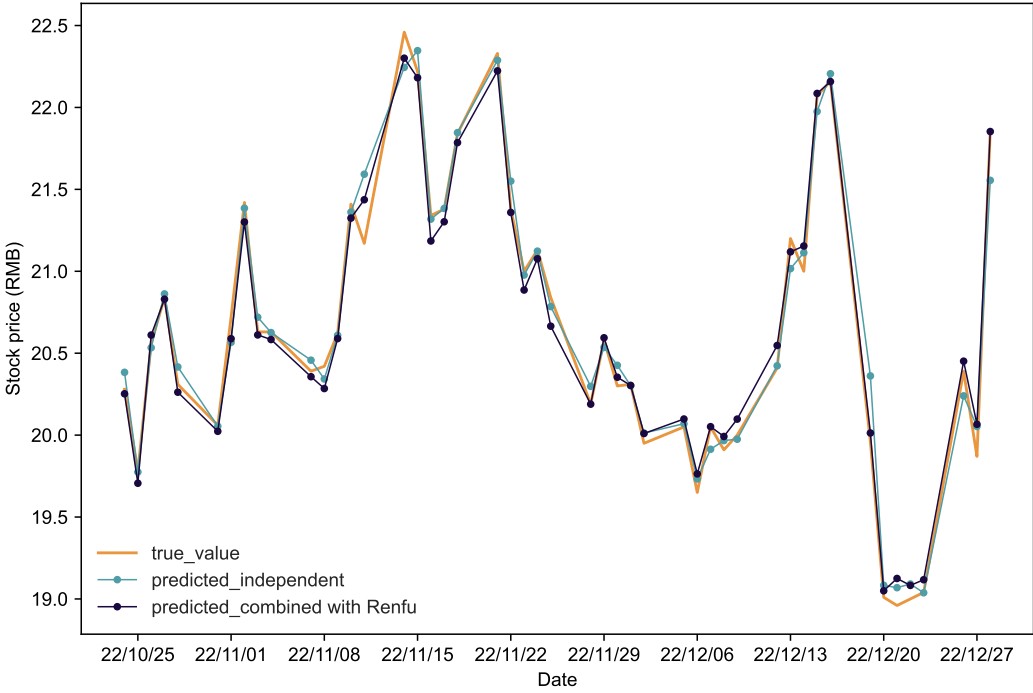

**Figure 5** **Comparative analysis of IPSO-LSTM model validation for Huahai stock price forecasting with and without interdependence.** True value is depicted by the orange line, predicted results without stock interdependence are represented by the blue line, and predicted results with stock interdependence taken into account are illustrated by the black line.

significantly correlated stocks. Target stock data, along with weighted data from non-target stocks in the same sub-industry *via* PSM, were merged for IPSO-LSTM model validation. The findings indicate that incorporating non-target stock data through PSM enhances the predictive accuracy of target stock prices. This underscores the positive role of PSM in stock price prediction and its capacity to improve accuracy.

Our study significantly advances stock market prediction models by addressing a crucial gap related to interdependencies among stocks within the same industry. Building on recognized literature gaps, our results confirm and extend the understanding of these interdependencies. Introducing and applying the PSM, our research surpasses previous studies by capturing the impact of industry-specific relationships on stock prices. The unique contribution lies in our method's application to study stock interdependence, an underutilized perspective in existing literature. Additionally, our integration of IPSO with LSTM networks for parameter optimization represents a groundbreaking advancement. While literature hints at the importance of parameter optimization, our study boldly explores and applies IPSO with LSTM networks for stock price prediction within specific industries. This integration not only fills a potential research gap but also offers a practical and innovative methodology to enhance predictive accuracy in stock market analysis. In summary, our research offers a novel perspective on stock interdependence, and

introducing an innovative approach for parameter optimization in machine learning models.

Our research advances stock market research on theoretical, managerial, and practical fronts. Theoretically, we introduce the PSM method to study stock interdependencies, enhancing the understanding of relationships between stocks. Additionally, our integration of IPSO with LSTM networks provides a novel approach, advancing machine learning in stock market research. On a managerial level, our findings offer practical insights for stock market professionals. Recognizing and incorporating interdependencies among industry stocks informs more accurate decision strategies, empowering managers to navigate market trends effectively. Practically, our research introduces advanced statistical and machine learning methods, specifically PSM and IPSO with LSTM networks, for improved stock price predictions in the pharmaceutical industry. These applications provide tangible tools, empowering analysts, researchers, and stakeholders with techniques to enhance predictive accuracy and optimize decision-making.

While this study has achieved notable advancements, there exist areas that warrant further refinement and enhancement. Firstly, this study has not considered real-world factors in the financial markets. Subsequent research endeavors will aim to incorporate these real-world factors into the model, enabling a more profound exploration of their substantive impact on stock interdependence and price predictions. This will ultimately result in the provision of more comprehensive and precise stock price forecasts. Secondly, the current methodology employs a single sliding window. Future work could benefit from the exploration of sliding windows of varying scales applied to stock price sequences, thereby enriching the predictive model. Finally, the research has primarily focused on investigating the interdependencies among stocks within the same sub-industry, without delving into the correlations between stocks from different sub-industries. Future endeavors could leverage PSM to unveil associations across different sub-industries, thereby creating more comprehensive matching datasets. These proposed improvements have the potential to contribute to a more thorough and robust understanding of stock price prediction and its underlying factors.

In considering future research directions, several avenues can extend the current findings. Firstly, exploring the application of PSM across diverse industries could illuminate its generalizability. Secondly, delving into the impact of additional variables, such as macroeconomic indicators, on the relationship between stocks within the same sub-industry could yield a more comprehensive understanding. Additionally, future studies may explore the implications of PSM for machine learning models beyond the IPSO-LSTM approach. Comparing the effectiveness of alternative methodologies could enrich the toolkit available to researchers and practitioners in the field. Lastly, investigating the real-time applicability of PSM in stock price prediction and its responsiveness to market changes could provide valuable insights for timely decision-making. These suggested directions collectively offer a broad and comprehensive perspective for further exploration in the dynamic and evolving field of stock market research.

In conclusion, this study demonstrates that integrating non-target stock data from the same sub-industry through PSM substantially enhances predictive accuracy, underscoring

its positive influence on stock price prediction. This research marks a pioneering use of PSM in studying stock interdependence, conducts a thorough examination of effects within the pharmaceutical industry, and leverages the IPSO optimization algorithm to boost LSTM network performance. These contributions offer a fresh perspective on stock price prediction research.

### Funding
The authors received no funding for this work.

### Competing Interests
The authors declare there are no competing interests.

### Author Contributions
- Shengnan Li conceived and designed the experiments, performed the experiments, analyzed the data, performed the computation work, prepared figures and/or tables, and approved the final draft.
- Lei Xue conceived and designed the experiments, authored or reviewed drafts of the article, and approved the final draft.

### Data Availability
The original data utilized in this study was retrieved from the Tongdaxin financial terminal. It includes stock data pertaining to the pharmaceutical sector, precisely segmented into chemical pharmaceuticals, biopharmaceuticals, and traditional Chinese medicine.

The dataset and accompanying source code (the dataset employed for IPSO-LSTM prediction of stock values) are available at Figshare: L, sn (2023). data and code. figshare. Dataset. https://doi.org/10.6084/m9.figshare.24115149.v1

### Supplemental Information
Supplemental information for this article can be found online at http://dx.doi.org/10.7717/peerj-cs.1819#supplemental-information.

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
