# Peer review of "The application of the propensity score matching method in stock prediction among stocks within the same industry"

_PeerJ Computer Science, doi:10.7717/peerj-cs.1819_

## Round 0.1 · original submission · Major Revisions

Dear authors,

Your paper has been reviewed by two reviewers who asked for revisions of the paper. Please revise the paper according to comments by reviewers, mark all changes in new version of the paper and provide cover letter with replies to them point to point.

**Language Note:** The review process has identified that the English language must be improved. PeerJ can provide language editing services - please contact us at copyediting@peerj.com for pricing (be sure to provide your manuscript number and title). Alternatively, you should make your own arrangements to improve the language quality and provide details in your response letter. – PeerJ Staff

Reviewer 1 ·

Basic reporting

When making stock price predictions (as in your study) there are some different alternative methods that can be used to focus on the relationships between stocks (which include Machine Learning Models, Co-integration Models, Vector Error Correction Models, Multivariate Regression Models, Stock Indices and Sector Analysis, Time Series Analysis, Stock Correlation Matrix Analysis etc.). These methods can increase your forecast accuracy and provide you with a more comprehensive stock-based marketing analysis. Sure, each method has some advantages and disadvantages. Which method is best may vary depending on your purpose, data, and needs.

Also, there are various models and algorithms you can use to make stock price predictions (ARIMA, GARCH, Markov Chain Monte Carlo, Fama-French Three Factor Model, Bayesian Networks, and Deep Learning Models). Optimization algorithms can be used to tune model parameters to achieve the best performance (Random Search, Genetic Algorithms, Bayesian Optimization, etc.). In your study, you used the Propensity Score Matching (PSM) method to investigate the relationships between stocks and the Improved Particle Swarm Optimization (IPSO) algorithm to improve performance by optimizing parameters. So you claimed that stock price prediction accuracy has increased significantly.

What are the advantages and disadvantages of the method and model algorithm you propose compared to the previous ones (methods and models )? On what basis did you make your choice? Would you like to compare the model you use with one of these? I would also like to learn your convincing arguments supported by literature and rationality regarding your model.

Experimental design

What are the advantages and disadvantages of the method and model algorithm you propose compared to the previous ones (methods and models )? On what basis did you make your choice? Would you like to compare the model you use with one of these? I would also like to learn your convincing arguments supported by literature and rationality regarding your model.

Validity of the findings

In your study, you used the Propensity Score Matching (PSM) method to investigate the relationships between stocks and the Improved Particle Swarm Optimization (IPSO) algorithm to improve performance by optimizing parameters. So you claimed that stock price prediction accuracy has increased significantly.

What are the advantages and disadvantages of the method and model algorithm you propose compared to the previous ones (methods and models )?

Reviewer 2 ·

Basic reporting

1. English is acceptable, although there are few minor style errors.
2. Introduction is not written well. It is mostly a literature review. Introduction should provide information on background, main aim, research questions, used methodology, main conclusions and contributions. I suggest the authors to move most of what is currently in introduction into a separate section "Literature review" and write another introduction.
3. Literature review should be expanded and updated. The authors should cover all aspects of the paper subject with an appropriate literature. In addition, only three references are from the last 3 years. Try to add some more up-to-date references.
4. The authors did not identify any research gaps based on the reviewed literature. This should be added.
5. Captions for some figures and tables are too extensive. They should be concise and informative. Please shorten them as much as possible. If you need to explain them, do it in the main text.

Experimental design

6. The subject of the paper is in line with the aims and scope of the Journal, although this connection should be made more clearly throughout the paper.
7. The authors did not clearly define research questions. Please establish them in the introduction and discuss them in the discussion.
8. The authors should check the notation. All symbols used in the equations must be defined. Some of them are currently not defined (e.g. "P"). Check the rest of them.
9. The authors should make it more clear what exactly is novel in their methodology. They say it's novel, but does not state the novelties explicitly.

Validity of the findings

10. The paper does not have a proper discussion. Authors did not discuss how the results can be interpreted in perspective of previous studies. Discussion should clearly and concisely explain the significance of the obtained results in order to demonstrate the actual contribution of the article to this field of research, when compared with the existing and studied literature.
11. The authors did not provide theoretical (managerial) and practical implications of the study.
12. Research questions should be shortly discussed in the conclusion.
13. The authors should try to add a few more (more general) future research directions.

Additional comments

14. The authors mentioned in the text a work by "Rosenbaum and Rubin, 1983." but did not list it in the reference list. All references from the paper must be present in the reference list and vice versa. Check the rest of the references.

---

## Round 0.2 · Minor Revisions

Dear authors,

Please revise the paper according to the comments of reviewer 1.

Reviewer 1 ·

Basic reporting

The authors' efforts and responses to further this work appear satisfactory. I don't have many objections. However, it is useful to highlight some critical factors about the realities of financial markets, which I have highlighted below.

Yes, detecting when different stocks act similarly or oppositely can provide valuable information. It is necessary to further mention some important issues regarding the interdependence effects between stocks and their importance in stock price forecasts, which can be listed as diversification, risk management, market trends, market sensitivity, cointegration, stationarity, and macroeconomic factors. Understanding how these factors affect different stocks helps make more accurate predictions. Moreover, taking into account a number of factors that disrupt interdependence (such as geopolitical events or changes in market sentiment) is important to improve the overall forecast accuracy in stock markets. I recommend that the topics mentioned here be discussed more throughout the article and especially in the literature section.

Experimental design

no comment

Validity of the findings

no comment

Additional comments

no comment

Reviewer 2 ·

Basic reporting

The authors have successfully addressed all issues from the previous review round.

Experimental design

The authors have successfully addressed all issues from the previous review round.

Validity of the findings

The authors have successfully addressed all issues from the previous review round.

Additional comments

The authors have successfully addressed all issues from the previous review round.

---

## Round 0.3 · accepted · Accept

Dear authors,

Your performed corrections and reply to the reviewer in the last version of the paper are satisfactory so that the paper can be accepted.